# Co-Infection of Potential Tick-Borne Pathogens of the Order Rickettsiales and *Borrelia burgdorferi* s. l. and Their Link to Season and Area in Germany

**DOI:** 10.3390/microorganisms11010157

**Published:** 2023-01-07

**Authors:** Angeline Hoffmann, Thomas Müller, Volker Fingerle, Cornelia Silaghi, Matthias Noll

**Affiliations:** 1Institute for Bioanalysis, Department of Applied Sciences, Coburg University of Applied Sciences and Arts, 96450 Coburg, Germany; 2Synlab Medical Care Unit, Department of Molecular biology, Tick Laboratory, 92637 Weiden in der Oberpfalz, Germany; 3Bavarian Health and Food Safety Authority (LGL), National Reference Center for Borrelia, 85764 Oberschleißheim, Germany; 4Institute of Infectology, Friedrich-Loeffler-Institute, Federal Research Institute of Animal Health, 17493 Greifswald, Germany; 5Bayreuth Center of Ecology and Environmental Research (BayCEER), University of Bayreuth, 95440 Bayreuth, Germany

**Keywords:** *Borrelia*, Rickettsiales, tick-borne pathogens, co-infection

## Abstract

The prevalence of potential human pathogenic members of the order Rickettsiales differs between *Borrelia burgdorferi* sensu lato-positive and -negative tick microbiomes. Here, co-infection of members of the order Rickettsiales, such as *Rickettsia* spp., *Anaplasma phagocytophilum*, *Wolbachia pipientis*, and *Neoehrlichia mikurensis* as well as *B. burgdorferi* s.l. in the tick microbiome was addressed. This study used conventional PCRs to investigate the diversity and prevalence of the before-mentioned bacteria in 760 nucleic acid extracts of *I. ricinus* ticks detached from humans, which were previously tested for *B. burgdorferi* s.l.. A *gltA* gene-based amplicon sequencing approach was performed to identify *Rickettsia* species. The prevalence of *Rickettsia* spp. (16.7%, *n* = 127) and *W. pipientis* (15.9%, *n* = 121) were similar, while *A. phagocytophilum* was found in 2.8% (*n* = 21) and *N. mikurensis* in 0.1% (*n* = 1) of all ticks. Co-infection of *B. burgdorferi* s. l. with *Rickettsia* spp. was most frequent. The *gltA* gene sequencing indicated that *Rickettsia helvetica* was the dominant *Rickettsia* species in tick microbiomes. Moreover, *R, monacensis* and *R. raoultii* were correlated with autumn and area south, respectively, and a negative *B. burgdorferi* s. l. finding. Almost every fifth tick carried DNA of at least two of the human pathogenic bacteria studied here.

## 1. Introduction

In Germany, up to 24 tick species, which belong to seven different genera, are present. Nevertheless, *Ixodes ricinus* remains the most widespread tick species in Germany [1,2,3] and represents the most important vector for tick-borne pathogens [4,5]. Bacterial tick-borne pathogens cause infectious diseases such as Lyme borreliosis or rickettsioses and threaten human and animal health worldwide. The order Rickettsiales contain a variety of tick-borne pathogens, which can cause neoehrlichiosis (*Neoehrlichia mikurensis*), anaplasmosis (e.g., *Anaplasma phagocytophilum*), or rickettsioses (e.g., *Rickettsia aeschlimanniii*) [6,7,8]. Therefore, knowledge of their distribution in ticks is essential for risk assessment and disease prevention [9].

*Rickettsia* spp. are transmitted both transstadially and transovarially in a vector population, and ticks, fleas, lice, and mites, are their main reservoir [10,11]. The genus *Rickettsia* comprises at least 30 obligate intracellular Gram-negative species, more than half being recognized or potential human pathogens [12,13]. These species are classified into four groups: spotted fever group, epidemic typhus group, the *Rickettsia bellii* group, and the *Rickettsia canadensis* group [13,14]. In addition to endosymbiotic *Rickettsia* species, the human pathogenic species *R. raoultii, R. massiliae, R. slovaca, R. aeschlimannii, R. monacensis*, and *R. helvetica* can be assigned to the spotted fever group. The most common *Rickettsia* species in *I. ricinus* is *R. helvetica*, and its potential pathogenicity is increasingly discussed [15]. For some *Rickettsia* species vertebrates such as birds (*R. aeschlimannii* and *R. helvetica*), reptiles (*R. helvetica* and *R. monacensis*), and mammals (*R. helvetica* and *R. monacensis*) have also been described as potential hosts [13,15,16].

Another tick-borne pathogen of the order Rickettsiales is the obligate intracellular species *A. phagocytophilum* [17], which in Europe uses *I. ricinus* as a vector and, among others, deer as reservoir hosts [18]. As *A. phagocytophilum* is exclusively transstadially transmitted, the presence of a competent vertebrate host is necessary to maintain the natural cycle [18]. Strains of *A. phagocytophilum* are classified as either pathogenic, less-pathogenic or non-pathogenic [5,17]. Human granulocytic anaplasmosis in Europe has been occasionally reported [19].

Recently, the first successful isolation of *N. mikurensis* from clinical specimens was reported [20,21]. It is now also known that mainly small mammals serve as reservoir hosts and most likely *I. ricinus* as a vector for Germany [21,22].

*W. pipientis* are symbionts localized either as a commensal or mutualist in the Malpighian tubules and/or ovaries of ticks [23]. No human infection with *W. pipientis* has been reported, although it is consistently found in *I. ricinus.*

The risk of simultaneous transmission of different tick-borne pathogens, as well as the risk of co-infection in humans from a tick bite, has already been addressed [5,24]. Co-infection of *A. phagocytophilum* with *Rickettsia* spp. (2.2% [17]), *B. burgdorferi* s. l. (12.3% [25]), or *Babesia microti* (1.8% [26]) in *I. ricinus* was documented.

To investigate the diversity and the quantity of tick-borne pathogens in the tick microbiome, various molecular methods, including next-generation sequencing (NGS), have been used [27]. Microbiome members may interact within the tick and with each other either commensally or mutualistically [27]. Sometimes 16S rRNA gene-based surveys cannot provide a sufficient phylogenetic resolution below genus level [28]. Therefore, for example, protein-coding genes have also been employed to identify members of the genus *Rickettsia* (e.g., *gltA* or *ompA/C*) [29,30]. The *gltA* gene codes for citrate synthase, which catalyzes the synthesis of isocitrate from oxaloacetate and acetyl coA, the first step in the Krebs cycle [31]. Heterogeneity of *gltA* gene sequences from tick-associated and human pathogenic *Rickettsia* species has a divergence of 1% to 4%. In addition, the *ompA* and *ompC* genes have been used to assess variability within *Rickettsia* species [32]. Both genes encode for a major outer-membrane protein and are relevant genes for immunogenicity.

The goal of this study was to determine (I) the (co-)prevalence of *Rickettsia* spp., *A. phagocytophilum*, *W. pipientis*, and/or *N. mikurensis* in *I. ricinus* tick nucleic acid extracts, and (II) whether the variables such as *B. burgdorferi* s. l. finding, season/month, or area of the collection significantly affected the bacteria’s (co-)prevalence.

## 2. Materials and Methods

### 2.1. Ethics Statements

No formal approval was required for this study because we used existing tick nucleic acid extracts.

### 2.2. Nucleic Acid Extracts of Ticks and Pool Composition

In total, 2029 *I. ricinus* ticks without the differentiation of developmental stages were sent by physicians or directly by clients (human host) to the accredited tick laboratory of Synlab Medical Care Centre (MVZ) Weiden from January to December of 2018 (Appendix A). Subsequently, nucleic acids of each tick were extracted by Synlab MVZ, as described [33], to analyze the tick-borne pathogens (e.g., species of *B. burgdorferi* s. l., *Rickettsia* spp., and tick-borne encephalitis) requested by the physicians or client. After nucleic acid extraction, the presence of different *B. burgdorferi* s. l. strains were tested by a real-time TaqMan PCR targeting the *ospA* gene with the primer pair (5′-AATATTTATTGGGAATAGGTCTAA-3’ and 5′-CACCAGGCAAATCTACTGA-3’) and the probe (tm-FA TTAATAGCATGYAAGCAAAATGTTAGCA) as outlined earlier [34].

Based on the 2029 ticks sent to Synlab MVZ Weiden, a map of Germany was created using the associated postal code as described earlier [34] to illustrate the origin of the ticks (Appendix A). A total of 760 tick extracts, consisting of 50 µL each, were selected arbitrarily based on the variables of area, season, month, and *B. burgdorferi* s. l findings and were used for a more detailed analyses (Table 1 and Appendix A) and stored at −80°C until further use.

Subsequently, 2 µL of nucleic acid extract each from 10 ticks were pooled, sorted in each case according to *B. burgdorferi* s. l. finding (negative or positive), area, and month of submission date. However, since we already received extracted DNA from ticks, no morphological information could be considered in this study. Thus, 76 individual tick nucleic acid extract pools (760 ticks) were created, of which 38 pools were *B. burgdorferi* s. l.-positive, and 38 pools were *B. burgdorferi* s. l.-negative (Appendix A).

### 2.3. Protein-Coding Gene-Based PCR to Screen for Bacteria of the Order *Rickettsiales*

The pools of the nucleic acid extracts of ticks were screened for *A. phagocytophilum, N. mikurensis, Rickettsia* spp., and *W. pipientis* by conventional PCR. For a 50 µL reaction, 10 µL 5× HF buffer (Thermo Fisher Scientific, Waltham, MA, USA), 1 µL dNTP Mix [10 mM] (Thermo Fisher Scientific), 0.5 µL Phusion high-fidelity polymerase (Thermo Fisher Scientific), 10 µL 5× PCR-Enhancer (Biozym Scientific GmbH, Hessisch Oldendorf, Germany), 5 µL respective primer pairs (details see Table 2), 16.5 µL UltraPure™ DNase/RNase-free distilled water (Thermo Fisher Scientific) and 2 µL (5 to 75 ng genomic DNA) for each tick nucleic acid extract were applied per PCR reaction. UltraPure™ water (Thermo Fisher Scientific) was used as negative control. Nucleic acid extracts of *A. phagocytophilum* (extracted from *I. ricinus*), *N. mikurensis* (extracted from *Apodemus agrarius*), *R. helvetica* (extracted from *I. ricinus*), or *W. pipientis* (extracted from *Culex pipiens* laboratory colony) were used as positive controls. The PCR thermal profiles of *W. pipientis*, *Rickettsia* spp., or *A*. *phagocytophilum* specific PCR reactions were an initial denaturation at 98 °C for 2 min followed by 35 cycles of denaturation at 98 °C for 10 s, annealing at 56.2 °C (wsp-81F/wsp-691R), 54.3 °C (RH314/RH654) or 64.3 °C (ApMSP2f/ApMSP2r) for 30 s, extension at 72 °C for 30 s, and the final elongation at 72 °C for 10 min in a T100™ Thermal Cycler (Bio-Rad Laboratories GmbH, Feldkirchen, Germany). For *N. mikurensis*, the thermal profile of PCR had an initial denaturation at 98 °C for 2 min followed by 55 cycles of denaturation at 98 °C for 15 s, annealing at 59 °C (NM-128s/NM-1152as) for 30 s, extension at 72 °C for 45 s, and the final elongation at 72 °C for 10 min in a T100™ Thermal Cycler (Bio-Rad Laboratories GmbH). PCR amplicons were checked on a 3% agarose gel stained with ethidium bromide and visualized by UV light (Genoplex, VWR International GmbH, Darmstadt, Germany). Whenever a nucleic acid extract pool was positive for one of the respective species, the individual tick extracts were tested for the respective pathogen(s). Tick extracts that tested positive for *Rickettsia* spp. were sent to LGC Genomics GmbH (Berlin, Germany) for Illumina MiSeq paired-end sequencing using the same PCR primer pairs (RH314/RH654, Table 2) as sequencing primers. Raw bioinformatic data pre-processing was carried out as recently reported [33].

### 2.4. gltA Gene-Based Sequence Analyses

For the *gltA* gene-based sequencing data analysis, a reference database was created by downloading 13 *gltA* gene sequences described within the genus *Rickettsia* and transmitted by ticks, fleas, or mites from the National Center for Biotechnology Information (NCBI) web-based database (https://www.ncbi.nlm.nih.gov/; last data update for this study was carried out 27 December 2022) [14,39]. The obtained *gltA* gene sequences were trimmed at the *gltA* primer pair sequences (Table 2) and were used for an alignment by using the MEGA X software version 10.2.3 [40] and the MUSCLE algorithm [41].

The tick-borne *gltA* gene sequences were assembled and aligned to the *gltA* sequence database on the Galaxy training platform [42,43]. As Galaxy is used so far as the 16S rRNA gene pipeline, a few steps were modified to analyze *gltA* gene sequences. Data cleanup was conducted with a maximum sequence length of 325 bp. Sequences that were poorly aligned between the start- and endpoint of the *gltA* gene sequence, including homopolymers with a length greater than or equal to six bases, were removed from the dataset. Thereafter sequence alignment was pre-clustered to assemble sequences that were nearly identical to each other. Sequences with a difference of one in 100 bases potentially represent sequencing errors and not biological variability (here, three out of 300 bp). According to the Galaxy training platform, after removing chimeras, OTU clustering was performed on the obtained sequences [43].

OTUs were sorted in decreasing order according to the number of *gltA* gene-based amplicon sequence reads. Subsequently, all OTUs with a read number less than three reads (singletons and doubletons) were removed from the dataset. An OTU count table with 24 OTUs was obtained (Appendix A. Based on the final alignment a DNA-based phylogenetic tree was calculated by the neighbor-joining method using the Tamura–Nei model in MEGA X [40]. In total, 1397 positions in the final alignment were used for tree construction.

### 2.5. Statistics

The PCR results of each tick nucleic acid extract were analysed using the Fisher exact test with Bonferroni correction for each individual comparison of prevalence with other variables (*B. burgdorferi* s. l. finding, area, month, season, (Table 1)) to determine, which variable affected the tick-borne pathogen prevalence significantly. A Fisher exact test with Bonferroni correction was also performed for examining the prevalence of *Rickettsia* associated OTUs in nucleic acid extracts as revealed by the *gltA* gene-based amplicon sequencing data. All tests were performed with the R software version 3.5.2. The significance level for the Fisher exact test was set as α = 0.05. All statistical analyses were performed by RStudio, and Fisher exact test data were illustrated using Origin 2017 (OriginLab Corporation, Northampton, MA, USA).

## 3. Results

### 3.1. Rickettsia spp. and W. pipientis Showed Highest Prevalence in Tick Extracts

Among the 760 tick nucleic acid extracts tested, 16.7% were *Rickettsia* spp. positive, and 15.9% were *W. pipientis* positive. In addition, 2.8% of the 760 ticks were positive for *A. phagocytophilum* and 0.1% for *N. mikurensis*. The prevalence of *Rickettsia spp.* and *W. pipientis* was similar comparing *B. burgdorferi* s. l.-positive (*n* = 380) and *B. burgdorferi* s. l.-negative (*n* = 380) tick nucleic acid extracts (Table 3).

Furthermore, 62 (8.1%) tick nucleic acid extracts were characterized with more than one member of the order Rickettsiales or with additional co-infection of *B. burgdorferi* s. l., of which two (*n* = 60/7.9%) or three (*n* = 2/0.3%) tick-borne pathogens were found simultaneously (Table 4).

In particular, a high ratio of co-infection of *B. burgdorferi* s. l. and *Rickettsia* spp. or *W. pipientis* was identified, and a significant effect between the co-infection of *B. burgdorferi* s. l. and *Rickettsia* spp. (*p* = 4.15 × 10^−2^) as well as *W. pipientis* (*p* = 4.71 × 10^−2^) was observed (Figure 1 and Table 4). *B. burgdorferi* s. l. with *Rickettsia* spp. (64.9%) and *B. burgdorferi* s. l. with *W. pipientis* (36.4%) co-occurred particularly frequently in the area of Weiden i. d. Oberpfalz (*n* = 242) and Neustadt a.d. Waldnaab (*n* = 229), respectively, as part of the southeast area (Appendix A).

The variables of area, month, and season affected the prevalence of respective potential tick-borne pathogens of the order Rickettsiales in the tick nucleic acid extracts. There was a significantly higher prevalence of *W. pipientis* in the area southeast compared to the areas south and west. Further, the prevalence of *W. pipientis* was affected by season (highest in spring) or month (highest in May), while *A. phagocytophilum* and *Rickettsia* spp. showed no significance against any of the variables investigated (Figure 1).

### 3.2. Rickettsia-Positive Ticks were Dominated by R. helvetica

The *gltA* gene was sequenced from 127 *Rickettsia*-positive nucleic acid extracts by Illumina MiSeq paired-end sequencing, and 24 different OTUs remained after filtering procedures. As a result of blasting the sequences of the OTUs, five different *Rickettsia* species were identified (Figure 2 and Appendix A), and *R. helvetica* dominated 20 out of the 24 OTUs (94.4% of 2805 total sequence reads). Subsequently prevalent were *R. aeschlimannii* and *R. yenbekshikazakhensis*, respectively, each with two OTUs (OTU 2 and OTU 8, both 2.7%), *R. raoultii* with one OTU (OTU 3, 1.8%), and *R. monacensis* with one OTU (OTU 4, 1.0%) (Figure 2 and Appendix A).

### 3.3. Prevalence of Rickettsia Species was Affected by B. burgdorferi s. l. Co-Infection

The identified *Rickettsia* species were then tested for significance to the presence of *B. burgdorferi* s. l., the variables area, month and season (Figure 3). However, no association with any environmental variable was detected for *R. aeschlimannii*, but a significant effect was identified with *B. burgdorferi* s. l. (*p* = 0.011). For *R. helvetica*, on the other hand, no significant effect was found for any of the environmental variables tested.

## 4. Discussion

### 4.1. Co-Infection of Potential Tick-Borne Pathogens Are Season Dependent

This study used conventional PCRs to investigate the diversity and prevalence of *Rickettsia* spp., *A. phagocytophilum*, *W. pipientis*, and *N. mikurensis* in 760 nucleic acid extracts of *I. ricinus* ticks detached from humans, which were previously tested for the presence of *B. burgdorferi* s.l.. Results were correlated to the variables season, area, as well as *B. burgdorferi* s. l. prevalence of each tick extract via Fisher exact test. The prevalence of *Rickettsia* spp. in ticks varies widely within Europe (0.5–66%) and also within Germany (1–30.4%), depending on the infected host species [4,8]. Thus, the prevalence of *Rickettsia* spp. determined here is 16.7%, which is within the expected range [44,45,46]. In contrast, as described here in the study, a similarly high prevalence of *W. pipientis* has not yet been described in literature. Evidence suggests that these endosymbionts significantly impact their vectors’ fitness, reproduction, immunity, and other characteristics [24,47]. Thus, the viability of the tick itself and tick-borne pathogens appears to be dependent on *Wolbachia* [24,38]. In addition, using the Fisher exact test with a Bonferroni correction, a significant difference was found in the prevalence of *W. pipientis* in the variables of season (spring (14.0%) and autumn (12.4%)), and month (May (14.0%) and July (24.8%) or rather September (12.4%)) (Figure 1). The reason can be explained by the tick infection with *W. pipientis* as the ticks are usually parasitized by the wasp *Ixodiphagus hookeri*, which carries *Wolbachia* [38,48]. Since the wasps lay their eggs in *I. ricinus* in summer and the development of *I. hookeri* continues only with the blood meal of the nymph in summer, an increased prevalence of *W. pipientis* in autumn and a flattening of the *Wolbachia* infection over the winter months is plausible [48].

The prevalence of *A. phagocytophilum*, on the other hand, has been adequately described and is reported to have a prevalence of 1–17.4% in I. ricinus ticks in Germany [8]. The 2.8% prevalence detected here is thus low. According to Svitalkova et al., the developmental stage of the tick itself influences the prevalence of *A. phagocytophilum* [49]. They observed a high prevalence of *A. phagocytophilum,* especially in adult *I. ricinus*, whereas the prevalence in nymphs was low [49]. However, since no morphological data on the ticks sent in were prepared by the physicians or the diagnostic laboratory, it can only be assumed that the ticks sent in were nymphs rather than adults.

The simultaneous prevalence of two or three potential tick-borne pathogens in 7.9% (*n* = 760) of the examined ticks indicates a potential contact of the ticks to a broad spectrum of pathogens and/or hosts (Table 4) [44]. Klitgaard et al. summarized that double infection occurs in 1–22% of the examined *I. ricinus* ticks in Europe, and *Borrelia* spp. is involved in most co-infections [50]. A high co-infection of *B. burgdorferi* s. l. and *Rickettsia* spp. or *W. pipientis* were also shown here (Table 4), supported by a significant effect (Figure. 1). Concerning *Borrelia* spp. and *Rickettsia* spp., a similarly high co-infection [45], as well as a positive association [4], have already been described in the literature.

Our prevalence of 0.4% for *B. burgdorferi* s. l. and *A. phagocytophilum* co-infection is in line with other studies from Germany [45]. However, a higher rate could have been expected, as it is known that *A. phagocytophilum* increases the colonization ability of *B*. *burgdorferi* s. l. and leads to more severe disease courses with diverse disease manifestations through immunosuppressive effects [16,50]. However, since a relatively low *A. phagocytophilum* prevalence was obtained here, the detected co-infection seems logical.

It should be mentioned that the observed co-infections here should also be considered as purely incidental. First, nucleic acid extracts from ticks infected with *B. burgdorferi* s. l. were directly detected, and second, the prevalence of each pathogen was significantly higher as a co-infection, therefore we consider an interaction between the pathogens likely.

### 4.2. Prevalence of R. helvetica to Environmental Variables

*I. ricinus* is dominated by *R. helvetica* in Europe and Germany [4,15] compared to other continents such as North America, where *R. rickettsii* is the predominant species of the genus *Rickettsia* [13]. In addition, 94.4% of the *gltA* gene sequencing reads of *Rickettsia*-positive ticks were identical to sequences of *R. helvetica* (Figure 2; Appendix A). Since *I. ricinus* simultaneously serves as a vector of *R. helvetica*, and additional highly efficient transovarial transmission is very likely here, *R. helvetica* can be permanently maintained in the tick life cycle [8,17]. Additionally, *Rickettsia* spp. can be passed on to the eggs during fertilization by male ticks [17]. The high reservoir competence of diverse vertebrate hosts, such as deer, mice, or hedgehogs ensures geographic distribution and availability in diverse habitats [15]. Assignment of OTU 2 and OTU 8 is ambiguous as both OTUs matched 100% to *R. aeschlimannii* and Ca. R. yenbekshikazakhensis (Appendix A). Turebekov et al. described how *R. aeschlimannii* and Ca. R. yenbekshikazakhensis carry identical *gltA* gene sequences. Therefore, further investigations on these species in the future should be aimed at completely excluding the prevalence of Ca. R. yenbekshikazakhensis in Germany. Nevertheless, the presence of Ca. R. yenbekshikazakhensis in Germany is considered very improbable as it has so far only been found in Kazakhstan [51]. Moreover, the associated tick species *Haemaphysalis punctata* was identified as a probable vector for Ca. R. yenbekshikazakhensis, and this tick species is not part of the natural German tick diversity today [51,52]. However, *R. aeschlimannii* has occasionally been detected before in the microbiome of *I. ricinus* but was found to be transmitted by ticks of the genus *Hyalomma* in the African region [53,54]. An prevalence of these *Rickettsia* species in Germany is nevertheless possible, as *R. aeschlimannii* infected *Hyalomma* ticks have already been collected from several migratory birds, which occur in Germany and elsewhere [15]. *R. monacensis* in 1.0% of the reads seems realistic, as the species is distributed on one side throughout Europe and associated with *I. ricinus* [8,44,54]. However, the host’s risk of *R. monacensis* manifesting infection can be considered low, as only four clinical cases have been described recently [54].

Thus, while *I. ricinus* mainly harbours the *Rickettsia* species *R. helvetica*, *R. aeschlimannii*, and *R. monacensis* [8], the presence of *R. raoultii* in the dataset (Figure 2, Appendix A) is novel. *R. raoultii* has already been described in Germany [15] but only connected with *D. reticulatus* and *D. marginatum* [15,17]. *R. raoultii* was present in 2.2% of the examined ticks. As *R. helvetica* and *R. monacensis* are mainly associated with *I. ricinus*, *R. raoultii* is strongly connected to *Dermacentor* ticks. As species identification was not carried out, the identified *R. raoultii* must likely have been harboured by an individual belonging to the genus *Dermacentor*. Therefore, molecular biological confirmation should be performed in the future to confirm the morphological identification of the tick. Yet, a general seasonality of *Rickettsia* spp. has been observed in several studies [17,46,55,56], which we also found (Figure 1 and Section 3.1). The results of the Fisher exact test for the *Rickettsia* species support our assumption that *Rickettsia* species are adapted to the environmental variables as they revealed no dependencies with respect to the variables tested in this study (Figure 3). Moreover, the problem of PCR inhibition is crucial for both PCR and sequencing approaches, which have already been observed in the tick microbiome studies [57]. Therefore, upcoming studies should be aware of such PCR drawbacks.

## 5. Conclusions

The risk of co-infection of several potential tick-borne pathogens in tick microbiomes was high in the studied areas of Germany. Almost every fifth tick collected from humans carried at least two of the (potential) pathogens or genus (*Rickettsia* spp.) containing human pathogenic species investigated here. In addition, a few rarer rickettsial species (*R. monacensis*, *R. raoultii*) were identified as not established in the vector *I. ricinus*, suggesting that they use other vectors and live in smaller ecological niches. Furthermore, Ca. R. yenbekshikazakhensis was found, which is not considered to be established in Germany, or the vector *I. ricinus*, which requires further investigation. In contrast to the other human pathogenic species of the genus, the predominant species in Germany, *R. helvetica*, seems to have found better mechanisms to establish itself in *I. ricinus* and its hosts, thus ensuring its survival.

Nevertheless, the lack of morphological determination of the tick in the diagnostic laboratory must be considered here. In the future, this should be integrated into the analysis process. Only based on the data, such as species, sex, or stage, exclusion criteria of (potential) tick-borne pathogens can be created, which subsequently simplifies the evaluation of critical test results with very high probability.

## Figures and Tables

**Figure 1 microorganisms-11-00157-f001:**
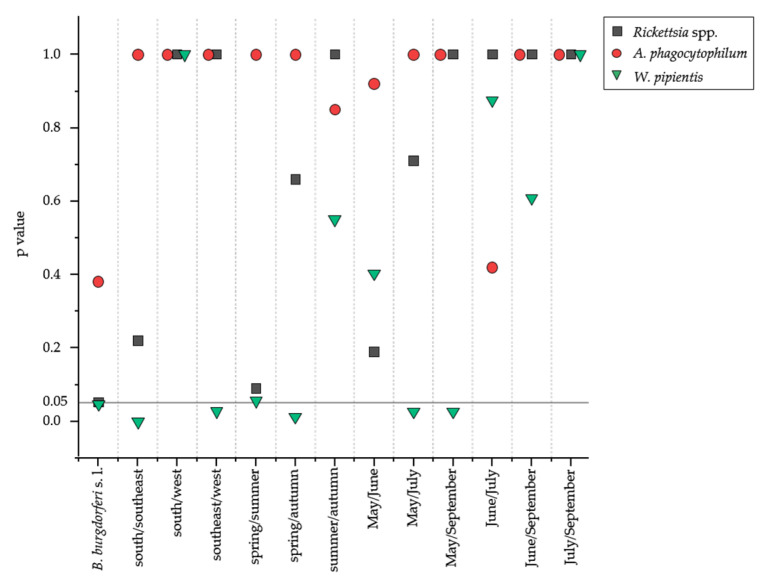
Effect of environmental variables on the prevalence of *Rickettsia* spp., *A. phagocytophilum*, *N. mikurensis*, and *W. pipientis* as revealed by the Fisher exact test with Bonferroni correction. Symbol legend for each species is included in the figure. Details of the characteristics of the variables can be found in Table 1. The significance level of 0.05 is indicated.

**Figure 2 microorganisms-11-00157-f002:**
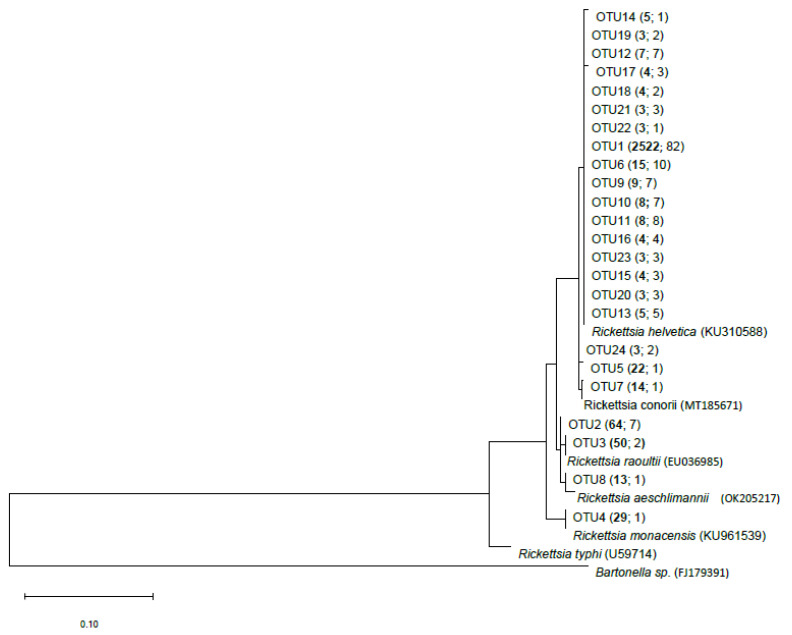
DNA-based neighbor-joining tree of partial *gltA* gene sequences. Environmental sequences retrieved from this study are indicated as OTU with their total sequence read numbers in bold and frequency of OTUs both in brackets. The *gltA* gene sequences from reference organisms were retrieved from GenBank and their accession numbers are indicated in brackets. The scale bar represents 0.1 nucleotide substitutions per site. For a sample-specific overview, see Appendix A.

**Figure 3 microorganisms-11-00157-f003:**
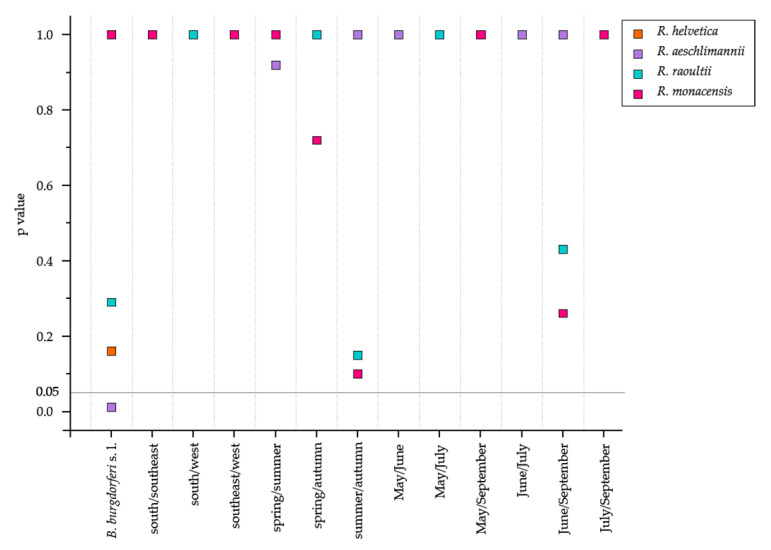
Effect of environmental variables on the prevalence of *R. helvetica*, *R. aeschlimannii*, *R. raoultii*, and *R. monacensis* as revealed by the Fisher exact test with Bonferroni correction. Symbol legend for each species is included in the figure. Details of the characteristics of the variables can be found in Table 1. The significance level of 0.05 is indicated.

**Table 1 microorganisms-11-00157-t001:** Characteristics of the variables of each tick nucleic acid extract for subsequent Fisher exact test. Seasons were categorized according to the meteorological calendar (spring: 1 March, summer: 1 June, autumn: 1 September); *n* indicates how many tick nucleic acid extracts were available per characteristic.

Variable	Characteristic
*B. burgdorferi* s. l.	Positive (*n* = 380), negative (*n* = 380)
Area	South (*n* = 120), southeast (*n* = 560), west (*n* = 80)
Month	May (*n* = 180), June (*n* = 380), July (*n* = 140), September (*n* = 60)
Season	Spring (*n* = 180), summer (*n* = 520), autumn (*n* = 60)

**Table 2 microorganisms-11-00157-t002:** Primer pairs used in this study to detect species of the order Rickettsiales in tick nucleic acid extracts.

Bacterial Species/ Genus	Target Gene	Amplicon Length [bp]	Final Primer Concentration (nM)	Primer Set *	References
*A. phagocytophilum*	*msp2*	77	400	ApMSP2f: 5′-ATGGAAGGTAGTGTTGGTTATGGTATT-3′ ApMSP2r: 5′-TTGGTCTTGAAGCGCTCGTA-3′	[35]
*N. mikurensis*	*groEL*	1024	500	NM-128f: 5′-AACAGGTGAAACACTAGATAAGTCCAT-3′ NM-1152f: 5′-TTCTACTTTGAACATTTGAAGAATTACTAT-3′	[36]
*Rickettsia* spp.	*gltA*	340	400	RH314f: 5′-AAACAGGTTGCTCATCATTC-3′ RH654r: 5′-AGAGCATTTTTTATTATTGG-3′	[37]
*W. pipientis*	*wsp*	591	500	wsp-81f: 5′-TGGTCCAATAAGTGATGAAGAAAC-3′ wsp-691r: 5′-AAAAATTAAACGCTACTCCA-3′	[38]

* f, forward and r, reverse. Primers were named as introduced in the respective reference.

**Table 3 microorganisms-11-00157-t003:** Presence of *A. phagocytophilum*, *N. mikurensis*, *Rickettsia* spp., and *W. pipientis* in individual tick nucleic acid extracts with *B. burgdorferi* s. l.-negative (*n* = 380) and *B. burgdorferi* s. l.-positive (*n* = 380) findings.

Bacterial Species or Genus	*B. burgdorferi* s. l. Finding	Prevalence of Positive Findings	Total Prevalence of Positive Findings
*A. phagocytophilum*	negative	13 (1.7%)	21 (2.8%)
	positive	8 (1.0%)	
*N. mikurensis*	negative	1 (0.1%)	1 (0.1%)
	positive	0 (0.0%)	
*Rickettsia* spp.	negative	53 (7.0%)	127 (16.7%)
	positive	74 (9.7%)	
*W. pipientis*	negative	50 (6.6%)	121 (15.9%)
	positive	71 (9.3%)	

**Table 4 microorganisms-11-00157-t004:** Co-infection of *B. burgdorferi* s. l., *A. phagocytophilum*, *N. mikurensis* and/or *Rickettsia* spp. of individual nucleic acid extracts from ticks. Prevalence is indicated by numbers and their relative prevalence in percentage (*n* = 760).

Frequence of co-Infection	Genus/ Species	Prevalence	Total Prevalence
Double	*B. burgdorferi* s. l. *+ A. phagocytophilum*	3/0.4%	60 (7.9%)
	*B. burgdorferi* s. l. *+ Rickettsia spp.*	57/7.5%	
Triple	*B. burgdorferi* s. l. *+ A. Phagocytophilum + Rickettsia spp.*	2/0.3%	2/0.3%
			62/8.1%

## Data Availability

The *gltA* gene sequences for tick samples were deposited in the NCBI nucleotide sequence databases under accession PRJNA839573.

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
