# Peer review of "Co-Infection of Potential Tick-Borne Pathogens of the Order Rickettsiales and *Borrelia burgdorferi* s. l. and Their Link to Season and Area in Germany"

_microorganisms, 2023, doi:10.3390/microorganisms11010157_

Round 1

Reviewer 1 Report

Good presentation. Only two minor suggestions: to indicate the stage(s) of ticks examined (line 102) and to check the italic writing of species and generic names.

Author Response

Response: First of all, thank you very much for your valuable time considering our manuscript as well as for your constructive comments and suggestions. We now addressed all comments and suggestions from both reviewers in our revised version of the manuscript with tracked changes as docx file. The point by point response to Reviewers are provided below.

Good presentation. Only two minor suggestions: to indicate the stage(s) of ticks examined (line 102) and to check the italic writing of species and generic names.

Response: Thank you for your tip. The developmental stages of analyzed ticks was not recorded in this study. To clarify this limitation, we already made a statement in the discussion section. However, to make this limitation clearer we added this information also in the material and method section (first sentence in the second paragraph 2.2) of the revised version of the manuscript.

All species names are now written in italics in the revised version of the manuscript.

Reviewer 2 Report

This manuscript is a useful contribution to the somewhat understudied matter of pathogen coinfection in ticks. The data is, as far as the reviewer can judge, sound and the laboratory methods appropriate, although a simpler sequencing method could with advantage have been employed.

The abstract and introduction are well-written and clear, although there is room for improvement, which is detailed below. 

The materials and methods and results, however, particularly the latter, are over-long and over-complicated, with the result that a simple and interesting set of results very difficult to understand. 

The discussion is rather long and tends to devote too much text to expatiating on one explanation for the findings while neglecting others. These are preliminary findings and do not justify such detailed explanations. It is enough to name the factors that may lie behind the observed associations and give references to relevant models. The discussion also needs linguistic attention; among other things, words such as 'thus', 'therefore' and 'however' have been used rather arbitrarily, which blurs the logic of the authors' arguments. 

These findings deserve to be published, but the manuscript needs to be greatly simplified and shortened. This will make it accessible to a broader audience and improve the article's impact without any appreciable loss of information. 

Title and ff. 

I question the use of the term co-occurrence. Co-infection (with or without the hyphen) is the term usually used in the literature and unambiguously implies the presence of two pathogens in the same tick. Co-occurrence is less specific and might equally imply the presence of the two pathogens in two ticks from the same place (co-localisation). Although I cannot say that the use of co-occurrence is incorrect, I do feel that it detracts from the clarity of the text. 

Prevalence

The author use, variously, 'prevalence', 'prevalence rate', 'abundance' and 'occurrence' mostly to refer to the same thing. 'Prevalence' is the most specific of these terms and using it consistently would improve the readability of the text. 

Introduction

L34 ff. The acronym 'TBP' does not add to the clarity of the text. Using the phrase 'tick borne pathogen' would, in my opinion, be preferable. 

L42. Deleting 'therefore'; 'along with'; 'known to be' will give a better sentence. 

L44. '... more than half being recognised or potential human pathogens' would be a better phrasing.  

L46-48. Imprecise formulation. Would 'The spotted fever group includes both endosymbiotic Rickettsia species and human pathogenic species...' correctly convey the authors' meaning? 

L50. Delete 'several' (?) 

L51. 'have also ... hosts'; move to end of sentence. 

L53. Delete 'among others' 

L57-60. The first of these two sentences seems to be incomplete, and as a result 'Nonetheless' in the second sentence seems out of place. 

L63. End of line. Delete 'In turn' and start a new paragraph. 

L64. 'endosymbionts' ... 'symbiont' is a tautology. Replacing 'symbiont' with 'mutualist' is suggested. 

L67. 'simultaneous transmission'. Perhaps the possibility of acquiring a double infection from two different tick bites might also be considered. This is a very real possibility, particularly in wild animals, but also in people in heavily tick-infested places. 

L70-74. These two sentences could be removed without loss. 

L76. Delete 'with amplicon sequencing'. If the authors intend '16s metabarcoding' then this should be stated explicitly. 

L77. Delete 'These' 

L81 - 82. Ambiguous. Does this mean that Midichloria and Rickettsia were the most abundant/prevalent bacterial species in I. ricinus ticks, or that I. ricinus was the tick species with the highest abundance/prevalence of Midichloria and Rickettsia

83. '...although it should be noted that these genera were described as endosymbionts, rather than pathogens.' would be better, although either way it seems to be hair-splitting and could be left out entirely. 

L85 (and elsewhere). I have not previously encountered the 'functional gene', as a synonym for 'protein-coding gene' before and I am not sure it is correct. Please check this and, if in doubt, use 'protein-coding gene'. 

86-88. I think this may be wrong. According to The Linkage Map of Escherichia K12, Edition 10 (Berlyn MKB, Microbial and Molecular Biology Reviews, 62: 814-894 (1998)), which I understand to be the paradigm for gene naming in bacteria, gltA codes for citrate synthase, which catalyses the synthesis of isocitrate from oxaloacetate and acetyl coA, the first step in the Krebs cycle. The large subunit of glutamate synthase, which catalyses the synthesis of glutamate from alpha-ketoglutarate and glutamine is coded by gltB. Please check this. 

L90. 'were' -> 'have been'; 'variabilities' -> 'variability'. 

L93. Delete 'Therefore'. 

Materials and methods 

Sections 2.2 and 2.3. The procedure for selection, pooling and testing of ticks is rather difficult to follow. I suggest using a flow chart or similar. Also please state an early point that ticks were screened in pools of ten, then tested individually if the pool was positive. 

L107. Please state the target gene for the Bbsl PCR. 

L112. Replace 'determine' with 'illustrate' (?)

L114. It is stated that tick extracts were "selected randomly base on (a set of variables)". A random selection is, by definition, not based on parameter. Random is a difficult word, as true randomization requires a randomization procedure. More correct terms are 'at random' or 'arbitrarily'.  

Section 2.3. I suggest describing the PCR programs in tabular form, either as an extension to Table 2 or in a separate table. PCR programs are not easy to describe in running text. 

L.134 and Table 2. This does not unambiguously define the final primer concentration in the PCR reaction mix. Were 5µl of primer at the (stock) concentration given in Table 2 added, or is the concentration given in Table 2 the final concentration? Please specify. 

L150. Remove commas. As written, this sentence implies a series of successive actions. 

L152. Delete 'was' or 'tested'. 

L152-3. "...the individual tick extract was tested for the respective pathogen(s)." would be better. 

Table 2 and sections 2.4 and 2.5. All taxonomic (species, genus etc.) names should be italicised. The same applies to gene names. 

L.180-185. This is very difficult to follow. Either clarify the text or remove it altogether. 

L.195-6. Delete "due to ... analysis". 

Results 

This section is excessively complicated.

In particular, the subsection on gltA sequencing makes a very complicated issue out of what is really a very simple problem. In the process, simple basic information like how many of the Rickettsia-positive ticks were species-identified becomes obscured by abstract terms like 'sequence motif' and OTU. In the meantime, the reason why sequencing failed for 103/127 Rickettsia-positive ticks is completely neglected. 

Similarly, CCA analysis seems an excessively advanced statistical technique to use on such a simple dataset. Indeed, it seems to add very little and purports to identify a significant correlation between R. monacensis and season even though Table 5 implies that only a single R. monacensis positive tick was found. 

L246-8. The analysis of the pool data does not seem to add anything, nor can I understand what purpose it serves when data for the individual ticks is available. 

This section can and should be drastically simplified. This is simple data and can be presented in a simple fashion and analysed with simple statistical techniques. 

Figure 3. 

This figure incorrectly refers to Table 2 for details of the variables. 

Discussion. 

The discussion seems unduly long and involved relative to the data. Although the explanations offered for seasonal effects are plausible, other explanations such as varying host availability and differential effects of climate on infected and uninfected ticks are also possible.

The same applies to the issue of co-occurrence. It is important to note that co-occurrence is on the whole only significant when its prevalence significantly exceeds the product of the prevalences of the individual pathogens. If one pathogen has a prevalence of 20% and the other has a prevalence of 10%, one expects a 2% prevalence of coinfection by chance alone. Coinfection is likely to be enhanced when the two pathogens have the same animal host and is not in itself an indication of interaction between the pathogens. 

Another important point to note is the effect of PCR inhibition. Inhibitory extracts are rather common when ticks are collected while feeding (see for example (Pedersen BN, Jenkins A and Kjelland V. Tick-borne pathogens in Ixodes ricinus ticks collected from migratory birds in southern Norway. PLoS ONE 15(4): e0230579). This tends to distort the observed prevalence of co-infection, as a tick in which one pathogen is detected is non-inhibitory and there is thus an a priori higher probability of detecting another pathogen. This effect will tend to be amplified by screening in pools, as one inhibitory tick may compromise the entire pool. In addition, inhibition may differentially effect different PCR reactions, thus distorting both actual and relative prevalences. As the authors have not controlled for inhibition in their material, this potential confounding factor should be acknowledged.  

Author Response

This manuscript is a useful contribution to the somewhat understudied matter of pathogen coinfection in ticks. The data is, as far as the reviewer can judge, sound and the laboratory methods appropriate, although a simpler sequencing method could with advantage have been employed.

The abstract and introduction are well-written and clear, although there is room for improvement, which is detailed below. 

The materials and methods and results, however, particularly the latter, are over-long and over-complicated, with the result that a simple and interesting set of results very difficult to understand. 

The discussion is rather long and tends to devote too much text to expatiating on one explanation for the findings while neglecting others. These are preliminary findings and do not justify such detailed explanations. It is enough to name the factors that may lie behind the observed associations and give references to relevant models. The discussion also needs linguistic attention; among other things, words such as 'thus', 'therefore' and 'however' have been used rather arbitrarily, which blurs the logic of the authors' arguments. 

These findings deserve to be published, but the manuscript needs to be greatly simplified and shortened. This will make it accessible to a broader audience and improve the article's impact without any appreciable loss of information. 

Response: First of all, thank you very much for your valuable time considering our manuscript as well as for your constructive comments and suggestions. We now addressed all comments and suggestions from both reviewers in our revised version of the manuscript with tracked changes as docx file. The point by point response to Reviewers are provided below.

Title and ff. 

I question the use of the term co-occurrence. Co-infection (with or without the hyphen) is the term usually used in the literature and unambiguously implies the presence of two pathogens in the same tick. Co-occurrence is less specific and might equally imply the presence of the two pathogens in two ticks from the same place (co-localisation). Although I cannot say that the use of co-occurrence is incorrect, I do feel that it detracts from the clarity of the text. 

Response: Thank you for the clarification. We changed according to the reviewer suggestion from co-occurrence to co-infection throughout in the revised version of the manuscript.

Prevalence

The author use, variously, 'prevalence', 'prevalence rate', 'abundance' and 'occurrence' mostly to refer to the same thing. 'Prevalence' is the most specific of these terms and using it consistently would improve the readability of the text. 

Response: Thank you for the clarification. We changed according to the reviewer suggestion to the wording prevalence throughout in the revised version of the manuscript.

Introduction

L34 ff. The acronym 'TBP' does not add to the clarity of the text. Using the phrase 'tick borne pathogen' would, in my opinion, be preferable. 

Response: We agree that this abbreviation is not essential and we changed according to the reviewer suggestion throughout in the revised version of the manuscript.

L42. Deleting 'therefore'; 'along with'; 'known to be' will give a better sentence. 

Response: Thank you, this change make the sentence more easy to read and we changed according to the reviewer suggestion in the revised version of the manuscript.

L44. '... more than half being recognised or potential human pathogens' would be a better phrasing.  

Response: We changed according to the reviewer comment.

L46-48. Imprecise formulation. Would 'The spotted fever group includes both endosymbiotic Rickettsia species and human pathogenic species...' correctly convey the authors' meaning? 

Response: Thank you for this critical reading and nice correction. We changed according to the reviewer comment.

L50. Delete 'several' (?) 

Response: We changed according to the reviewer comment.

L51. 'have also ... hosts'; move to end of sentence. 

Response: We changed according to the reviewer comment.

L53. Delete 'among others' 

Response: We changed according to the reviewer comment.

L57-60. The first of these two sentences seems to be incomplete, and as a result 'Nonetheless' in the second sentence seems out of place. 

Response: Thank you very much. We have rephrased these two sentences and now “nonetheless” should fit in the revised version of the manuscript.

L63. End of line. Delete 'In turn' and start a new paragraph. 

Response: We changed according to the reviewer comment.

L64. 'endosymbionts' ... 'symbiont' is a tautology. Replacing 'symbiont' with 'mutualist' is suggested. 

Response: We changed according to the reviewer comment.

L67. 'simultaneous transmission'. Perhaps the possibility of acquiring a double infection from two different tick bites might also be considered. This is a very real possibility, particularly in wild animals, but also in people in heavily tick-infested places. 

Response: Thank you for your comment. We argue that this is an obvious route of acquiring another microbial load. However, we do not know the tick history and the frequency of blood meals. As we do not want to extend the introduction, we prefer not to include this information in the manuscript just to be as concise as possible.

L70-74. These two sentences could be removed without loss. 

Response: Thank you for your comment and we agree that this information is optional. Therefore, we deleted this part according to the reviewer comment in the revised manuscript version.

L76. Delete 'with amplicon sequencing'. If the authors intend '16s metabarcoding' then this should be stated explicitly. 

Response: We changed according to the reviewer comment.

L77. Delete 'These' 

Response: We changed according to the reviewer comment.

L81 - 82. Ambiguous. Does this mean that Midichloria and Rickettsia were the most abundant/prevalent bacterial species in I. ricinus ticks, or that I. ricinus was the tick species with the highest abundance/prevalence of Midichloria and Rickettsia

Response: Thank you very much. We have rephrased this sentence to make our statement clearer in the revised version of the manuscript.

  1. '...although it should be noted that these genera were described as endosymbionts, rather than pathogens.' would be better, although either way it seems to be hair-splitting and could be left out entirely. 

Response: Thank you for your comment and we agree that this information is optional. Therefore, we deleted this part according to the reviewer comment in the revised manuscript version.

L85 (and elsewhere). I have not previously encountered the 'functional gene', as a synonym for 'protein-coding gene' before and I am not sure it is correct. Please check this and, if in doubt, use 'protein-coding gene'. 

Response: Good information. We found both option but like “protein-coding gene” more. Therefore, we exchanged functional gene to protein-coding gene according to the reviewer comment throughout the revised manuscript version (including captions).

86-88. I think this may be wrong. According to The Linkage Map of Escherichia K12, Edition 10 (Berlyn MKB, Microbial and Molecular Biology Reviews, 62: 814-894 (1998)), which I understand to be the paradigm for gene naming in bacteria, gltA codes for citrate synthase, which catalyses the synthesis of isocitrate from oxaloacetate and acetyl coA, the first step in the Krebs cycle. The large subunit of glutamate synthase, which catalyses the synthesis of glutamate from alpha-ketoglutarate and glutamine is coded by gltB. Please check this. 

Response: Thank you so much for careful reading and the information of the appropriate citation! Indeed, we made a mistake here, which has been corrected in the revised manuscript version according to the reviewer suggestion. We also added the citation.   

L90. 'were' -> 'have been'; 'variabilities' -> 'variability'. 

Response: We changed according to the reviewer comment.

L93. Delete 'Therefore'. 

Response: We changed according to the reviewer comment.

Materials and methods 

Sections 2.2 and 2.3. The procedure for selection, pooling and testing of ticks is rather difficult to follow. I suggest using a flow chart or similar. Also please state an early point that ticks were screened in pools of ten, then tested individually if the pool was positive. 

Response: Thank you for your comment. However, we think that our text clearly explains our experimental procedure. To meet the reviewers request we added a flow chart of the procedure as supplementary figure S1 in the revised version of the manuscript.

L107. Please state the target gene for the Bbsl PCR. 

Response: Thank you very much. We added the information that the OspA gene was the target gene in the revised manuscript version.

L112. Replace 'determine' with 'illustrate' (?)

Response: Thank you for your contribution to improve our manuscript. We changed according to the reviewer comment.

L114. It is stated that tick extracts were "selected randomly base on (a set of variables)". A random selection is, by definition, not based on parameter. Random is a difficult word, as true randomization requires a randomization procedure. More correct terms are 'at random' or 'arbitrarily'.  

Response: Thank you. We exchanged “randomly” to “arbitrarily” in the revised manuscript version.

Section 2.3. I suggest describing the PCR programs in tabular form, either as an extension to Table 2 or in a separate table. PCR programs are not easy to describe in running text. 

Response: Thank you for your contribution to improve. However, both ways to present this information can be found in the literature, and we think that our way is also easily accessible for the readership. Therefore, we do not change this part in the revised version of the manuscript.

L.134 and Table 2. This does not unambiguously define the final primer concentration in the PCR reaction mix. Were 5µl of primer at the (stock) concentration given in Table 2 added, or is the concentration given in Table 2 the final concentration? Please specify. 

Response: Thank you for your careful reading. We have exchanged in Table 2 the information “Molarity” by “final concentration” in the revised version of the manuscript to clarify this concentration for the readers.

L150. Remove commas. As written, this sentence implies a series of successive actions. 

Response: We changed according to the reviewer comment.

L152. Delete 'was' or 'tested'. 

Response: We changed according to the reviewer comment.

L152-3. "...the individual tick extract was tested for the respective pathogen(s)." would be better. 

Response: We changed according to the reviewer comment.

Table 2 and sections 2.4 and 2.5. All taxonomic (species, genus etc.) names should be italicised. The same applies to gene names. 

Response: We changed according to the reviewer comment.

L.180-185. This is very difficult to follow. Either clarify the text or remove it altogether. 

Response: Thank you for your comment and we agree that this information is optional. Therefore, we deleted this part according to the reviewer comment in the revised manuscript version.

L.195-6. Delete "due to ... analysis". 

Response: We changed according to the reviewer comment.

Results 

This section is excessively complicated.

In particular, the subsection on gltA sequencing makes a very complicated issue out of what is really a very simple problem. In the process, simple basic information like how many of the Rickettsia-positive ticks were species-identified becomes obscured by abstract terms like 'sequence motif' and OTU.

In the meantime, the reason why sequencing failed for 103/127 Rickettsia-positive ticks is completely neglected. 

Response: Thank you for this response. We sequenced a portion of the gltA gene from 127 Rickettsia-positive samples using Illumima-MiSeq technology. This gave us a total of 2805 different gltA gene sequences, which were distributed differently in terms of species (OTU) in the 127 Rickettsia-positive samples. After the usual filtering steps, a total of 24 different sequences (OTUs), each with a different read count, remained in 92 Rickettsia-positive samples. Therefore, we quantified the amount of Rickettsia-positive ticks and additionally assigned the respective OTUs to each tick extract to identify the respective Rickettsia species. This is essential to differentiate, if the same or dissimilar Rickettsia were found. We think that the need to sequence this gltA gene is not fully understood by the reviewer. As we believe that our text is explaining both the procedure and the results correctly we do not see any need to change here.

Similarly, CCA analysis seems an excessively advanced statistical technique to use on such a simple dataset. Indeed, it seems to add very little and purports to identify a significant correlation between R. monacensis and season even though Table 5 implies that only a single R. monacensis positive tick was found. 

Response: Thank you again for this information. We agree that CCA is not needed and we have deleted this information in the revised manuscript version. The statistical test using the Fischer exact test is retained in the text because we believe that this test is sufficient for comparing the effects. In addition, we modified the text blocks of the results at the Fischer exact test to ease the readerability.

L246-8. The analysis of the pool data does not seem to add anything, nor can I understand what purpose it serves when data for the individual ticks is available. 

Response: Thank you for your comment and we agree that this information is optional. Therefore, we deleted this part according to the reviewer comment in the revised manuscript version.

This section can and should be drastically simplified. This is simple data and can be presented in a simple fashion and analysed with simple statistical techniques.

Response: We have made many changes in this section of the revised manuscript version to simplify the presentation of our data and hope that our changes will satisfy the reviewer's criticism.

Figure 3. 

This figure incorrectly refers to Table 2 for details of the variables. 

Response: We changed according to the reviewer comment.

Discussion. 

The discussion seems unduly long and involved relative to the data. Although the explanations offered for seasonal effects are plausible, other explanations such as varying host availability and differential effects of climate on infected and uninfected ticks are also possible.

The same applies to the issue of co-occurrence. It is important to note that co-occurrence is on the whole only significant when its prevalence significantly exceeds the product of the prevalences of the individual pathogens. If one pathogen has a prevalence of 20% and the other has a prevalence of 10%, one expects a 2% prevalence of coinfection by chance alone. Coinfection is likely to be enhanced when the two pathogens have the same animal host and is not in itself an indication of interaction between the pathogens. 

Response: Thank you for this helpful hint. We added this information in the revised version of the discussion to meet the reviewer suggestion.

Another important point to note is the effect of PCR inhibition. Inhibitory extracts are rather common when ticks are collected while feeding (see for example (Pedersen BN, Jenkins A and Kjelland V. Tick-borne pathogens in Ixodes ricinus ticks collected from migratory birds in southern Norway. PLoS ONE 15(4): e0230579). This tends to distort the observed prevalence of co-infection, as a tick in which one pathogen is detected is non-inhibitory and there is thus an a priori higher probability of detecting another pathogen. This effect will tend to be amplified by screening in pools, as one inhibitory tick may compromise the entire pool. In addition, inhibition may differentially effect different PCR reactions, thus distorting both actual and relative prevalences. As the authors have not controlled for inhibition in their material, this potential confounding factor should be acknowledged.

Response: Thank you for this helpful hint. We added this information including the citation in the revised version of the discussion to meet the reviewer suggestion.

Round 2

Reviewer 2 Report

This manuscript is now much improved. Two issues remain, plus a few minor textual corrections, which are detailed below. 

(1) Statistical methods. I have discussed this with some of my colleagues who are better versed in statistics than myself. As far as we can ascertain, the authors have done Fisher exact tests for each individual comparison of prevalence with other variables. This should be stated explicitly. 

Unfortunately, doing this greatly increases the risk of type I error (false positive result): doing 40 tests with a significance limit of p=0.05 will almost guarantee some false positive results. It is our view that a Bonferroni correction should be applied, which in this example would entail reducing the significance limit to 0.05/40 = 0.00125. (See, for example, Whitlock and Schluter, The Analysis of Biological Data, 3rd Edition, 2020, pp. 459-461, or https://www.aaos.org/aaosnow/2012/apr/research/research7/) I would therefore suggest limiting the tests to bacteria where the numbers are sufficient to give statistically meaningful results in order to reduce the magnitude of the Bonferroni correction needed. 

I'm afraid this would nonetheless mean that many of the correlations that the authors have noted would no longer be significant. It will therefore be necessary to rewrite the discussion accordingly. Associations which do not reach the corrected significance level may still be noted as suggestive of a correlation, but only the briefest discussion of possible mechanisms is appropriate and then preferably where they confirm earlier findings. 

I apologise to the authors for not spotting this earlier. 

Section 3.2/Table 5. The OTU counts do not, in my view, add anything to the results. Rather, they tend to confuse them. What is of primary interest is how many tick specimens were ascribed to which Rickettsia species, and just as importantly, how many could not be ascribed to any species. 

The OTU data would be better presented as a phylogenetic tree. This would allow sequences to be assigned to a 'Rickettsia helvetic cluster', 'R. monacensis cluster', etc. Then a table showing how many samples mapped to each cluster can be provided. 

Minor textual comments: 

Abstract 

L18. differed -> differs 
L19/20. the coinfection of -> coinfection with 
L20. remove 'the genera'; 'the species'. 
L28/29. 'The gltA sequencing ... microbiomes'; better would be 'sequencing indicated that R. helvetica was the dominant Rickettsia species in the tick microbiomes'. 
L31/32. 'at least two bacteria ... effects'; better would be 'at least two of the human pathogenic bacteria studied here'. 

Results 
L35. Mover 'are present' to the end of the sentence. 
L59. 'competent hosts is obligate' -> 'competent vertebrate host is necessary'. 
L60/61. 'pathogenic, less pathogenic or non-pathogenic' would be a more logical order. 
L61. Delete 'Nonetheless'. 
L64. Delete 'In addition'.  
L66. Mutualism, commensalism (and parasitism) are different types of symbiosis. Therefore exchange 'symbiont' and mutualist' in the sentence:
'W. pipientis are symbionts localized, either as commensals or mutualists, in the Malpighian tubules...' 
L70/71. Delete 'For example'; 'of A. phagocytophilum with Rickettsia' -> 'with A. phagocytophilum and Rickettsia'. 
L76. Delete 'symbiotically'; see L66 above. Since mutualism is a type of symbiosis 'symbiotically or mutualistically' is a tautology. 
L78. were -> has been. 
L80. 'synthesis of isocitrate from oxaloacetate and acetyl CoA' 

Materials and methods. 
L107. I understand the authors' dilemma, as we are not supposed to begin a sentence with a number. Unfortunately, 'Thus' is not the right word. 'Consequently' or 'Then' would be better. Alternatively break the rules and begin the sentence with '760'. 
L146/147. extract was -> extracts were; delete 'in addition'. 
149. Move 'as reported recently' to the end of the sentence and invert the order: 'as recently reported'. 
Table 2. Header line. Sequence length -> Amplicon length; Final primer concentration, or move this column to the right to indicate that it refers to the primer, not the amplicon. 
2.4. I find the term 'outgroup' difficult to understand. The usual use of the term is in phylogenetic analysis where the outgroup is a distantly related species - and, by inference, one equally unrelated to all the other species, which is included in order to provide the tree with a root. Here, no phylogenetic analysis is mentioned, so the purpose of the outgroups is not clear to me. Nor is it necessarily the case that flea- and mite- borne Rickettsia and those not found in Europe are less related to those species found in ticks within Europe. As far as I can see the only purpose of these sequences is to improve the reliability of the sequence alignment. Thus, it would be better to say 'additional sequences from non-I. ricinus associated species (...species list) were included in the alignment.   

Author Response

This manuscript is now much improved. Two issues remain, plus a few minor textual corrections, which are detailed below. 

Response: We thank reviewer #2 very much for the constructive criticism of our manuscript, and we have corrected all addressed concerns in the revised manuscript version with tracked changes as docx file. The point by point response to Reviewers are provided below.

(1) Statistical methods. I have discussed this with some of my colleagues who are better versed in statistics than myself. As far as we can ascertain, the authors have done Fisher exact tests for each individual comparison of prevalence with other variables. This should be stated explicitly. 

Unfortunately, doing this greatly increases the risk of type I error (false positive result): doing 40 tests with a significance limit of p=0.05 will almost guarantee some false positive results. It is our view that a Bonferroni correction should be applied, which in this example would entail reducing the significance limit to 0.05/40 = 0.00125. (See, for example, Whitlock and Schluter, The Analysis of Biological Data, 3rd Edition, 2020, pp. 459-461, or https://www.aaos.org/aaosnow/2012/apr/research/research7/) I would therefore suggest limiting the tests to bacteria where the numbers are sufficient to give statistically meaningful results in order to reduce the magnitude of the Bonferroni correction needed. 

I'm afraid this would nonetheless mean that many of the correlations that the authors have noted would no longer be significant. It will therefore be necessary to rewrite the discussion accordingly. Associations which do not reach the corrected significance level may still be noted as suggestive of a correlation, but only the briefest discussion of possible mechanisms is appropriate and then preferably where they confirm earlier findings. 

I apologise to the authors for not spotting this earlier. 

Response: We thank reviewer #2 and his colleague as we have not used the Bonferroni correction in the previous manuscript version, but this correction is indeed correct for our dataset. Therefore, we have recalculated all statistical tests, changed Figs. 1 and 3, material & methods, results and discussion section in the revised manuscript version. 

Section 3.2/Table 5. The OTU counts do not, in my view, add anything to the results. Rather, they tend to confuse them. What is of primary interest is how many tick specimens were ascribed to which Rickettsia species, and just as importantly, how many could not be ascribed to any species. 

The OTU data would be better presented as a phylogenetic tree. This would allow sequences to be assigned to a 'Rickettsia helvetic cluster', 'R. monacensis cluster', etc. Then a table showing how many samples mapped to each cluster can be provided. 

Response: We thank reviewer #2 and very nice idea. We have constructed a neighbour joining tree, which replaced the former table 5. In addition, we added some more information in the material & methods section, and changed text blocks in the results and discussion section of the revised manuscript. 

Minor textual comments: 

Abstract 

L18. differed -> differs 
L19/20. the coinfection of -> coinfection with 
L20. remove 'the genera'; 'the species'. 
L28/29. 'The gltA sequencing ... microbiomes'; better would be 'sequencing indicated that R. helvetica was the dominant Rickettsia species in the tick microbiomes'. 
L31/32. 'at least two bacteria ... effects'; better would be 'at least two of the human pathogenic bacteria studied here'. 

Results 
L35. Mover 'are present' to the end of the sentence. 
L59. 'competent hosts is obligate' -> 'competent vertebrate host is necessary'. 
L60/61. 'pathogenic, less pathogenic or non-pathogenic' would be a more logical order. 
L61. Delete 'Nonetheless'. 
L64. Delete 'In addition'.  
L66. Mutualism, commensalism (and parasitism) are different types of symbiosis. Therefore exchange 'symbiont' and mutualist' in the sentence:
'W. pipientis are symbionts localized, either as commensals or mutualists, in the Malpighian tubules...' 
L70/71. Delete 'For example'; 'of A. phagocytophilum with Rickettsia' -> 'with A. phagocytophilum and Rickettsia'. 
L76. Delete 'symbiotically'; see L66 above. Since mutualism is a type of symbiosis 'symbiotically or mutualistically' is a tautology. 
L78. were -> has been. 
L80. 'synthesis of isocitrate from oxaloacetate and acetyl CoA' 

Materials and methods. 
L107. I understand the authors' dilemma, as we are not supposed to begin a sentence with a number. Unfortunately, 'Thus' is not the right word. 'Consequently' or 'Then' would be better. Alternatively break the rules and begin the sentence with '760'. 
L146/147. extract was -> extracts were; delete 'in addition'. 
149. Move 'as reported recently' to the end of the sentence and invert the order: 'as recently reported'. 
Table 2. Header line. Sequence length -> Amplicon length; Final primer concentration, or move this column to the right to indicate that it refers to the primer, not the amplicon. 

Response: We thank the reviewer for his careful reading and helpful corrections. All points of minor textual comments were corrected in the revised manuscript version.

2.4. I find the term 'outgroup' difficult to understand. The usual use of the term is in phylogenetic analysis where the outgroup is a distantly related species - and, by inference, one equally unrelated to all the other species, which is included in order to provide the tree with a root. Here, no phylogenetic analysis is mentioned, so the purpose of the outgroups is not clear to me. Nor is it necessarily the case that flea- and mite- borne Rickettsia and those not found in Europe are less related to those species found in ticks within Europe. As far as I can see the only purpose of these sequences is to improve the reliability of the sequence alignment. Thus, it would be better to say 'additional sequences from non-I. ricinus associated species (...species list) were included in the alignment.   

Response: Sorry for this confusion. Following our tree construction and associated changes in the text, this part of the text was also restructured. As a result, the mentioned text sections became obsolete and were thus deleted in the revised manuscript version.